Perch, Perca fluviatilis show a directional preference for, but do not increase attacks toward, prey in response to water-borne cortisol

Henderson Lindsay J. Lindsay.Henderson@newcastle.ac.uk 1 2
Ryan Mary R. 2
Rowland Hannah M. 3
1 Centre for Behaviour and Evolution, Newcastle University , Newcastle upon Tyne , United Kingdom
2 Institute of Biodiversity, Animal Health and Comparative Medicine, College of Medical, Veterinary & Life Sciences, The University of Glasgow , Glasgow , United Kingdom
3 Predators and Prey Research Group, Max Planck Institute for Chemical Ecology , Jena , Germany
Piato Angelo
Electronic publication date: 2017 Oct 3
Publication date: 2017
Volume: 5
Electronic Location ID: e3883
Received 2017 Apr 13; Accepted 2017 Sep 12
Copyright: ©2017 Henderson et al.
Copyright year: 2017
Copyright holder: Henderson et al.
License: This is an open access article distributed under the terms of the Creative Commons Attribution License, which permits unrestricted use, distribution, reproduction and adaptation in any medium and for any purpose provided that it is properly attributed. For attribution, the original author(s), title, publication source (PeerJ) and either DOI or URL of the article must be cited.
License URL: https://creativecommons.org/licenses/by/4.0/

Keywords: Sensory ecology, Gasterosteus aculeatus, Perca fluviatilis, Hormone, Predation

Funding: Association for the Study of Animal Behaviour Junior Research Fellowship from Churchill College Zoological Society of London This work was funded by an Association for the Study of Animal Behaviour grant. During the study and manuscript writing HMR was supported by a Junior Research Fellowship from Churchill College, Cambridge and an Institute Fellowship from the Zoological Society of London. The funders had no role in study design, data collection and analysis, decision to publish, or preparation of the manuscript.

==============================
In freshwater environments, chemosensory cues play an important role in predator-prey interactions. Prey use a variety of chemosensory cues to detect and avoid predators. However, whether predators use the chemical cues released by disturbed or stressed prey has received less attention. Here we tested the hypothesis that the disturbance cue cortisol, in conjunction with visual cues of prey, elevates predatory behavior. We presented predators (perch, Perca fluviatilis) with three chemosensory choice tests and recorded their location, orientation, and aggressive behavior. We compared the responses of predators when provided with (i) visual cues of prey only (two adjacent tanks containing sticklebacks); (ii) visual and natural chemical cues of prey vs. visual cues only; and (iii) visual cues of prey with cortisol vs. visual cues only. Perch spent a significantly higher proportion of time in proximity to prey, and orientated toward prey more, when presented with a cortisol stimulus plus visual cues, relative to presentations of visual and natural chemical cues of prey, or visual cues of prey only. There was a trend that perch directed a higher proportion of predatory behaviors (number of lunges) toward sticklebacks when presented with a cortisol stimulus plus visual cues, relative to the other chemosensory conditions. But they did not show a significant increase in total predatory behavior in response to cortisol. Therefore, it is not clear whether water-borne cortisol, in conjunction with visual cues of prey, affects predatory behavior. Our results provide evidence that cortisol could be a source of public information about prey state and/or disturbance, but further work is required to confirm this.

Introduction

The outcomes of predator–prey interactions are largely influenced by the ability of predators and prey to detect and respond to one another (Lima & Dill, 1990; Endler, 1991). An organism that perceives a predatory threat or potential prey before being detected itself gains an advantage (Lunt & Smee, 2015). Prey can successfully avoid predators when they have the sensory advantage by detecting and avoiding predators before being consumed (Mirza & Chivers, 2002). In contrast, predators prevail when they hold the perceptual advantage and detect prey before they are themselves detected and prey can escape. There is a built-in imbalance between predator and prey in regard to the penalty of failure during an encounter: failure for prey is death, whereas for a predator it is only a lost meal (Dawkins & Krebs, 1979). This asymmetry in the selective pressure on predators and prey, known as the ‘life-dinner’ principle (Dawkins & Krebs, 1979), is reflected in the greater research attention afforded to prey responses over predator behavior (Ferrari, Wisenden & Chivers, 2010). A greater understanding of how predators respond to cues released by prey could provide greater insight into the selective forces that shape the evolution of predator–prey interactions.

Predators and prey use cues across multiple sensory channels to detect one another, including visual, acoustic, chemical, electrical and/or tactile cues (Ferrari, Wisenden & Chivers, 2010). In aquatic systems chemical cues are a principal source of information (DeBose & Nevitt, 2008; Ferrari, Wisenden & Chivers, 2010). Prey increase anti-predator behaviors in response to the chemical cues of predators (e.g., kairomones Ferrari, Wisenden & Chivers, 2010), as well as the chemical cues released by prey that have been injured by predators (e.g., damage-released alarm cues, Mathis, Chivers & Smith, 1993; Harvey & Brown, 2004; Chivers, Brown & Ferrari, 2012; Lönnstedt, McCormick & Chivers, 2012), and by prey that have been disturbed or stressed by predators, without being injured or consumed (e.g., disturbance cues, Chivers, Brown & Smith, 1996; Vavrek et al., 2008; Vavrek & Brown, 2009; Ferrari, Wisenden & Chivers, 2010; Brown et al., 2012). By responding to the chemical cues of heterospecifics and conspecifics, about the presence of predators, prey have increased chances of survival during encounters with predators (Mirza & Chivers, 2002).

Predators detect and respond to damage-released alarm cues to localize prey (Mathis, Chivers & Smith, 1995; Wisenden & Thiel, 2002; Lönnstedt, McCormick & Chivers, 2012). While this suggests that the release of damage-released alarm cues would be detrimental for prey, research into the responses of predators have revealed additional fitness benefits to prey (Chivers, Brown & Smith, 1996; Lönnstedt & McCormick, 2015). Prey that release alarm cues after injury attract secondary predators, which increases their probability of surviviving the encounter by 35–40% (Lönnstedt & McCormick, 2015). Therefore, investigating the responses of predators can provide insight into the selective dynamics acting on both cues and the producers of cues (Mathis, Chivers & Smith, 1995; Chivers, Brown & Smith, 1996; Lönnstedt & McCormick, 2015).

In contrast, whether predators respond to disturbance cues released by prey has received less attention (Chivers, Brown & Ferrari, 2012). Disturbance cues are interesting because prey could have more control over their release (Chivers, Brown & Ferrari, 2012), in comparison to damage-released alarm cues that are released after injury. Disturbance cues could benefit both prey and predators if their release reliably informs predators that they have been detected and are therefore less likely to be successful in capturing prey (e.g., pursuit deterrent, Woodland, Jaafar & Knight, 1980; Caro, 1995). However, if predators can detect and use disturbance cues to localize prey, selection could act to reduce their release by prey in environments with a high density of predators. Therefore, disturbance cues are predicted to have more diverse selection pressures acting on them than damage-released alarm cues (Chivers, Brown & Ferrari, 2012).

Disturbance cues exist in a variety of systems, including invertebrates, amphibians, and freshwater fish (Hazlett, 1990; Wisenden, Chivers & Smith, 1995; Kiesecker et al., 1999; Bryer, Mirza & Chivers, 2001; Mirza & Chivers, 2002; Jordão, 2004; Brown et al., 2012). It has been suggested that nitrogenous compounds in urine, and respiratory byproducts could act as disturbance cues (Hazlett, 1990; Kiesecker et al., 1999; Vavrek & Brown, 2009; Brown et al., 2012). One potential disturbance cue that has received less research attention is the stress hormone cortisol (Olivotto et al., 2002; Vavrek & Brown, 2009), the principal glucocorticoid in teleost fish (Brown, Gardner & Braithwaite, 2005). Prey fish show an acute elevation of their circulating cortisol concentrations in response to the presence of predators (Breves & Specker, 2005; Barcellos et al., 2007). Cortisol can pass across the gills into the surrounding water (Scott & Ellis, 2007; Sebire, Katsiadaki & Scott, 2007), and can influence the physiological stress responses of conspecifics (Toa, Afonso & Iwama, 2004; Barcellos et al., 2011). Elevated cortisol concentrations are associated with reduced body mass and growth, as well as increased parasite loads in fish species (Jentoft et al., 2005; Fast et al., 2006). These factors in turn have been linked to reduced predator avoidance in prey (Sogard, 1997; Ness & Foster, 1999). It is therefore possible that cortisol could provide information about prey state and/or disturbance, and that predators could become attuned to this physiological response of prey. Whether predators can detect and respond to the presence of cortisol released by prey or use the presence of the hormone to inform their foraging decisions is unknown. This is an important gap in our knowledge because if predators can eavesdrop upon disturbance cues and use them to localize prey, this could influence the selective advantage of the release of disturbance cues by prey (Ferrari, Wisenden & Chivers, 2010).

To answer this question, we investigated whether predators respond specifically to cortisol, and/or other ‘signature’ chemical cues released by prey (Wyatt, 2010), when also presented with visual cues of prey. We used the piscivorous European perch (Perca fluviatilis) as model predator, and three-spined stickleback (Gasterosteus aculeatus) as prey. We presented each perch with three chemosensory prey choice tests (i) visual cues only of two adjacent tanks containing sticklebacks; (ii) visual and natural chemical cues of sticklebacks on one side vs. visual cues only on the other; and (iii) visual cues of sticklebacks with a cortisol stimulus on one side vs. visual cues only on the other (Fig. 1). We predicted that, (a) if predators use cortisol to identify and localize prey, predators should show a preference for prey associated with cortisol over prey without the chemical cue; or (b) if predators use cortisol as a signal of vigilant prey and reduced likelihood of attack success, predators should reduce their preference for prey with the cortisol cue. For both behavioral responses, if it is cortisol specifically that attracts predators we predict their response to be similar if cortisol is presented alone, or with additional chemical cues of sticklebacks. Alternatively, (c) our null hypothesis was that perch would not show a preference for prey when visual cues are augmented with the chemical cue cortisol.

Figure 1 Schematic of experimental tanks.

Pump system A allowed transfer of fresh water or cortisol solution into the perch tank. Pump system B allowed transfer of the natural chemical cues from the stickleback holding water.

Materials and Methods

All fish were collected from water-bodies in west central Scotland. Three-spined sticklebacks (n = 150, ∼1 g) were collected in October–November 2011 from Balmaha Pond, a small water body that does not contain piscivorous fish. Perch (∼16–20 cm length, 52.5 ± 9.8 g), a natural predator of sticklebacks (Gross, 1978), were collected in October 2010 and 2011 from Howend Trout Fishery. Perch were likely to have had experience of sticklebacks at Howend Trout Fishery. Fish were housed at indoor aquaria at the University of Glasgow, UK. Fish were fed daily with frozen bloodworm (Glycera dibranchiate), and kept under a light regime of 12L:12D. Mains tap water was used for holding water and water temperature was 16.5 ± 1 °C with a pH of 8–8.4. All experiments were conducted between December 2011 and January 2012, using a tank design that allows transfer of chemical and visual cues between fish (Le Vin, Mable & Arnold, 2010), while preventing physical interaction for ethical reasons (Fig. 1, Association for the Study of Animal Behaviour (ASAB), 2001).

Experimental procedure

Each perch was introduced into a 45 × 20 × 25 cm experimental tank ∼20 h before a test to allow for acclimation after handling, and to standardize hunger between individuals. Adjacent to the experimental tank on either side were two flow-through stimulus tanks (10 × 20 × 25 cm), each maintained with 2l of water. Five sticklebacks were introduced into each stimulus tank, also ∼20 h before a test. Flow-through stimulus tanks were used to avoid accumulation of cortisol in the water during acclimation, caused by capture and handling and due to the small size of tanks (Pottinger et al., 2011; Archard et al., 2012). Removable opaque barriers were positioned between the experimental and stimulus tanks to prevent the perch and sticklebacks from seeing each other during acclimation. These were removed at the start of a test to allow predators and prey to see one another.

To test the influence of hormonal cues on predator behavior, each perch participated in three chemosensory choice tests (number of perch = 18, total number of tests = 54): (i) a control test, where fresh water was dripped into the perch tank on both sides directly in front of the stimulus tanks holding sticklebacks (Fig. 1A); (ii) a natural chemical test, where water from one of the stimulus tanks containing naïve sticklebacks (no previous exposure to perch) was dripped into one side and fresh water dripped into the other (Figs. 1A and 1B); and (iii) a cortisol test, where we provided our putative disturbance cue at a concentration that a predator might encounter in nature (0.2 ng/ml of free cortisol; Sigma) on one side and fresh water on the other (Fig. 1A). Sticklebacks have been documented to release up to 0.26 ng/ml/h cortisol into holding water after capture and handling (Sebire, Katsiadaki & Scott, 2007; Pottinger et al., 2011). Because perch are diurnal visual predators, in all tests predators were given visual cues of sticklebacks. Visual cues were provided on both sides of the perch’s tank and only chemical cues changed between tests.

The order of chemosensory tests and the side placement of the chemical stimulus were randomized. Water was drawn from neighboring stimulus tanks through a pump system with a flow rate of 1.6 ml/min, which has been shown to induce a reaction to the chemical stimulus in other fish species (McLennan & Ryan, 1997; Le Vin, Mable & Arnold, 2010). A previous study has shown using color-dyed water, that water dripped into a focal tank remained mainly localized within the zone (13 × 8 × 19) immediately adjacent to where the water was dripped (Le Vin, Mable & Arnold, 2010).

In the natural chemical tests, sticklebacks had not been used previously in the experiment (naïve). Our rationale here was that they would show a natural stress response and release elevated levels of cortisol, after visual exposure to perch (Barcellos et al., 2007). Sticklebacks were in some cases re-used in the other two tests to reduce the total number of sticklebacks used in the experiment (in line with the 3Rs; Association for the Study of Animal Behaviour (ASAB), 2001). Sticklebacks were not used on consecutive days or more than twice a week, as per the terms of our Home Office licence. In a sub-sample of stickleback tanks (n = 17 tanks containing five sticklebacks; nine were naïve, eight were re-used) we found no difference in the time the sticklebacks spent in the side of the tank closet to the perch (GLM t = 1.09, P = 0.29), or the maximum number that nipped at perch (GLM t = 0.98, P = 0.33), between naïve and re-used tanks of sticklebacks.

At the start of each test the opaque barriers were removed, the stimulus pumps were started and the flow-through system in the stimulus tanks was stopped. We observed the perch for 60 min from behind a blind to avoid any effect of the observer on fish behavior. We visually divided the perch tank into three equal zones (15 cm length), two ‘preference zones’ (chemical stimulus zone and the opposite non-chemical stimulus zone) located adjacent to the front of each stimulus tank and a ‘non-preference’ zone in between them (Fig. 1). For analysis in the control tests where there were no chemical stimuli, the chemical stimulus zone was allocated to the left side in 50% of tests and to the right for the other 50%.

We used Observer 8 software (Noldus Information Technology, Wageningen, The Netherlands) to record the duration spent by perch in each zone; the duration spent orientating toward sticklebacks in each zone; the number of predatory attacks made toward the sticklebacks (hereon termed lunges, which were defined as rapid forward movements toward sticklebacks). We also counted the number of times fish moved between zones to quantify their level of activity. At the end of the 60 min the perch and sticklebacks were returned to their holding tanks, and the experimental tanks that held perch and sticklebacks were then drained and cleaned before their use in the next test. Each perch had a two day break between tests.

Cortisol concentration in experimental tanks

To assess the concentration of cortisol released by stickleback into their holding water we collected the water from their tanks in the natural chemical cue tests, 60 min after presentation of the perch (n = 9). To measure the level of cortisol released by perch into their holding water in the experimental tanks, we collected water after 60 mins when no stimulus water was being pumped (n = 4).

To determine the cortisol concentrations in the water samples, cortisol was extracted by pulling 100 ml under vacuum at a rate of ca. 5 ml/min through primed SPE cartridges containing octadecylsilane (Sep-pak Plus; Waters Ltd., Watford, UK). Water samples were not pre-filtered. Priming involved 5 ml methanol followed by 5 ml distilled water. After the samples had been pulled through, the cartridges were washed with 5 ml distilled water followed by 20 ml air (to remove as much moisture as possible). The free steroids (not conjugated Ellis et al., 2004) were eluted with 5 ml ethyl acetate. These were collected in a glass tube and evaporated at 45 °C under a stream of air. Samples were then re-suspended in 250 µl of assay buffer (Tris-buffered saline) and cortisol concentrations were established using a Cortisol ELISA Kit (Enzo Life Sciences, Inc., Farmingdale, NY, USA). Samples were run within two assays, all samples run in triplicate and recoveries calculated after a known amount of cortisol was added to samples were >96% and the mean intra-and inter-assay coefficient of variation were 8.6% and 8.4% respectively. 100 ml water samples were used because cortisol is found at low concentrations in holding water (Scott & Ellis, 2007), and this methodology allowed the amount of cortisol to be concentrated so that values could be measured on the linear phase of the standard curve (sample value range: 14.2–4.4 ng/ml, standard curve: 100–0.78 ng/ml). As the 100 ml water samples were re-suspended in 250 µl of buffer for the assay (×400 concentration), cortisol values from the assay were divided by 400 to give the concentration in the holding water (ng/ml).

Statistical analyses

To investigate whether perch behavior differed between the three chemosensory choice tests, we calculated the proportion of total time that perch spent in the chemical stimulus zone, the proportion of time that perch spent orientating toward the sticklebacks in the chemical stimulus zone, and the proportion of total lunges at sticklebacks in the chemical stimulus zone. We calculated proportions because we were comparing between choice tests. We calculated proportions as the time spent in the chemical stimulus zone divided by the sum of the time spent in all zones. We calculated the proportion of time orienting toward sticklebacks using the same method. For lunges at sticklebacks, we calculated the proportion of lunges in the chemical stimulus zone divided by the sum of lunges in the chemical stimulus zone and the opposite stimulus zone (visual cues only). All proportions were arcsine-transformed to normalize data.

Linear Mixed Models (LMMs) were used to assess whether chemosensory test type (control, natural cues, cortisol) influenced the proportion of time spent in the chemical stimulus zone, the proportion of time orienting in the chemical stimulus zone, and the proportion of lunges by perch at sticklebacks in the chemical stimulus zone. We also used a LMM to test whether the number of movements perch made around their tank differed between chemosensory tests (although these data were a count they were normally distributed). We used Generalized Linear Mixed Models (GLMMs) to investigate whether total lunges at sticklebacks differed between chemosensory tests with a negative binomial error structure (due to overdispersion). Perch identity was included as a random factor in all models to account for repeated measures. To control for multiple tests, we applied a Bonferroni correction to our level of statistical significance: effects were only considered significant if P < 0.01.

The order in which chemosensory tests were presented to perch did not influence their behavior (P > 0.70), and so this factor was not included in final models. Chemosensory test type did not affect whether perch began the test in the chemical stimulus zone (binomial GLMM z = 0.76, P = 0.45). We analyzed whether chemosensory test affected if perch were active or remained immobile during the 60 min observation using a binomial GLMM. We then split our analyses, to focus on actively behaving perch for all behavioral analysis (where perch showed at least one behavior: n = control, 16; natural cues, 15; cortisol, 11). We removed immobile individuals because at the start of each trial a perch could be located in any zone within the tank: if the focal perch did not move for the entire test, the perch would be assigned 100% of time to a single zone, and 100% of time orientating in a single direction. This would necessarily have skewed our data. For the proportion of lunges, we did not include perch that did not lunge, and thus contributed zeros (n = 6). This is because a zero value could be perceived incorrectly as 100% of lunging behavior occurring at the opposite stimulus zone (n = control, 14; natural cues, 13; cortisol, 9).

All analyses were conducted using the nlme package, and the lme4 package for GLMMs in R 2.12.2 (R Development Core Team, 2012). The data used for analyses can be found in Data S1.

Ethics statement

This study was performed under a UK Home Office licence (60/4292) and was subject to review by the University of Glasgow ethics committee. All experiments were conducted using a tank design that allows transfer of chemical and visual cues between fish, while preventing physical interaction for ethical reasons. The hormone concentrations used in this study are within the natural levels experienced by fish in the wild. Prey fish were provided with shelter while in visual contact with the predator. Permission to catch perch was granted by a privately-owned trout fishery (Howend) and sticklebacks were collected from Balmaha pond after obtaining permission from the Scottish government. This study did not involve endangered or protected species.

Results

Predator behavior

Of the 18 perch, nine did not move during an entire test (n = 12 out of 54 tests), and this was marginally, but non-significantly, more likely to occur in trials when cortisol was the chemical cue (binomial GLMM z = 1.82, P = 0.06).

When we focus on actively moving individuals, perch spent a significantly higher proportion of their time in the chemical stimulus zone when cortisol was added compared to natural chemical cues of sticklebacks and the water control (Table 1 & Fig. 2A). Perch also oriented toward sticklebacks for a significantly higher proportion of time in the chemical stimulus zone when cortisol was added compared to natural chemical cues of sticklebacks and the water control (Table 1 & Fig. 2B). There was a marginally non-significant trend for perch to show a higher proportion of lunges toward sticklebacks in the chemical stimulus zone when cortisol was added, compared to the water control (Table 1 & Fig. 2C). Perch did not lunge more overall at sticklebacks in the chemical stimulus zone during the cortisol tests compared to the other chemosensory tests (Fig. 3A, GLMM z =  − 0.35, P = 0.73). Perch moved around the tank equally frequently in each chemosensory test (Fig. 3B, LMM t = 0.38, P = 0.70).

Table 1 Table of LMMs results.

Linear Mixed Models testing the effect of chemical trial upon proportion of time in chemical stimulus zone, proportion of time orienting in chemical stimulus zone, and proportion of lunges in chemical stimulus zone. Individual ID was included as a random factor in all models. Values in bold denote statistically significant factors.

Treatments	β	s.e.	t	P	Random intercept	
					% σ	
Proportion of time in chemical stimulus zone	0.33	0.09	3.61	0.001	35.8	
Proportion of time orienting in chemical stimulus zone	0.62	0.16	3.93	<0.001	2.0e−03	
Proportion of lunges in chemical stimulus zone	0.32	0.17	1.93	0.07a	50.8	
Notes.

a Marginally non-significant factor.

Figure 2 Scatter dot plots showing how chemical cues influenced perch behavior.

Scatter dot plots showing median ± inter-quartile range of (A) the proportion of total time spent by perch in chemical stimulus zone, (B) the proportion of time spent by perch orientating toward sticklebacks in the chemical stimulus zone, and (C) the proportion of lunges at sticklebacks in the chemical stimulus zone.

Figure 3 Scatter dot plots showing how chemical cues influenced perch behavior.

Scatter dot plots showing median ± inter-quartile range of (A) the total lunges by perch at sticklebacks in the chemical stimulus zone, and (B) total movements between zones by perch. n = control, 16; stickleback natural cues, 15; cortisol, 11.

When we analyze perch behavior including only those perch that were active under all three treatments (nine perch, and 27 trials) our results are consistent with previous analysis. (Proportions: Duration, GLMM t = 3.62, P = 0.002; Orienting, GLMM t = 3.91, P < 0.001, Lunges, LMM t = 1.87, P = 0.08. Total lunges: GLMM z = 0.27, P = 0.78).

Concentration of cortisol in experimental tanks

The mean concentration of cortisol in the natural chemical tests was an order of magnitude lower than the cortisol test; 0.02 ng/ml (S.E. < 0.001) vs. 0.2 ng/ml, and was lower than the mean concentration of cortisol in the perch holding water; 0.05 ng/ml (S.E. < 0.001).

Discussion

We predicted that the disturbance cue cortisol would affect our predators’ behavior toward prey, manifesting as either an increased attraction to or avoidance of prey. Our results support both the attraction and deterrence hypothesis. We found that our predators spent more time in close proximity to prey, as well as more time oriented toward prey, when prey were associated with cortisol. To our knowledge, this is the first evidence that water-borne cortisol can increase interest toward prey. Our results are in line with the findings of other research that has shown predators are attracted to, and use damage-released alarm cues to detect prey (Chivers, Brown & Smith, 1996; Wisenden & Thiel, 2002; DeBose & Nevitt, 2008; Ferrari, Wisenden & Chivers, 2010; Elvidge & Brown, 2012; Lönnstedt, McCormick & Chivers, 2012). However, while there was a trend that perch showed a higher proportion of attacks toward prey in the cortisol test, perch did not show a statistically significant increase in total attacks toward prey in the cortisol tests. We also observed that some of our predators were immobile throughout a test, and that this was marginally more likely to occur in response to prey associated with cortisol. Therefore, it is unclear whether cortisol is strictly involved in predatory behavior. The observed immobility could be a deterrence effect of cortisol, i.e., that perch perceived cortisol as a cue of prey vigilance (Woodland, Jaafar & Knight, 1980; Smith, 1992). Alternatively, immobility could be an antipredator response by our predators.

Responding to cortisol may aid predators in localizing prey. In addition, higher levels of cortisol associated with prey may indicate that they have a reduced ability to avoid predators, as elevated cortisol has been found in fish in poorer condition (Sogard, 1997; Ness & Foster, 1999). Alternatively, avoidance of cortisol by predators could be advantageous if it indicates that they have been detected and prey are vigilant (Woodland, Jaafar & Knight, 1980), or that other predators are foraging in an area (Olsson & Brown, 2006). The presence of cortisol could also indicate a predator’s own predation risk. Predators may respond differently to chemical cues of prey depending upon their life-history stage or prior experience. For example, in yellow perch (Perca flavescens) there is an ontogenetic shift from insectivory to piscivory, and this is associated with contrasting responses to the skin extracts of conspecifics: adult perch increase their foraging behavior in response to the skin extracts of conspecifics, while juveniles elevate their anti-predator behavior (Harvey & Brown, 2004). Our results do not provide unequivocal evidence that water-borne cortisol specifically influences predatory behavior. As wild caught perch were used, we were unable to control for their prior experience before capture. This could have influenced their behavioral response to the chemical cue cortisol. To further understand the behavioral responses of predators to the disturbance cue cortisol, it may be important to use predatory fish that are reared in captivity to control for experience. Furthermore, research into the behavioral responses of non-predatory fish to cortisol, and the response of predators to cortisol in the presence of conspecifics are required. This would confirm that behavioral changes are predatory and not a general response to cortisol.

A limitation of our study is that the sticklebacks we used as prey released less cortisol into their holding water following visual exposure to a predator than sticklebacks that have experienced handling stress (Sebire, Katsiadaki & Scott, 2007; Pottinger et al., 2011). This is likely because our sticklebacks originated from a water body that did not contain piscivorous fish. Many prey species do not show innate recognition of potential predators (Mathis, Chivers & Smith, 1993; Chivers & Smith, 1994), so a lack of experience of predatory fish may explain the low level of stress response. Also, prey species often require chemical and/or mechanical stimuli in addition to visual cues to respond to the presence of a predator (Stauffer & Semlitsch, 1993; Breves & Specker, 2005; Barcellos et al., 2007), which could further explain our findings. We suggest that the lower level of cortisol in the natural cues test is why our predators did not show a difference in their behavior in the natural chemical test compared to the water control. But due to this limitation, we were unable to examine whether perch showed a preference for visual cues of sticklebacks associated with both elevated cortisol and natural “signature” chemical cues (Wyatt, 2010). We could have added cortisol to the natural cues test, but as we had anticipated that the sticklebacks in the natural cues test would show a cortisol response after visual exposure to the perch (Barcellos et al., 2007), we assumed that doing this would have elevated cortisol to ecologically irrelevant high levels (>0.46 ng/ml). Despite this limitation, we did show that predators respond to cortisol over the natural chemical cues (e.g., nitrogenous compounds in urine, and respiratory by-products) the prey fish released while visually exposed to predators. Future experiments with predator-experienced prey would be worthwhile.

We also did not test whether perch showed a behavioral response to chemical cues alone. As perch are sight-dependent diurnal predators, they likely require both visual and chemical cues to target prey (Turesson & Brönmark, 2004; Elvidge & Brown, 2012). Although chemical-cue-only tests could have provided information on the relative importance of visual and chemical cues, our design still allowed us to investigate how chemical cues augment the behavior of predators in comparison to visual cues only.

If predators only respond to cortisol when it rises above the background levels of cortisol, as proposed in the background noise hypothesis (Vavrek & Brown, 2009), then we would only have detected behavioral changes in predators if sticklebacks produced more than the background level of 0.05 ng/ml. The concentration used in our cortisol tests, was higher than the background levels and was within the range released by sticklebacks after a handling stress (Sebire, Katsiadaki & Scott, 2007; Pottinger et al., 2011). Therefore, this concentration could be experienced by perch in the wild, so should be biologically meaningful (Sebire, Katsiadaki & Scott, 2007; Pottinger et al., 2011). In a real world setting, cortisol is likely to be ubiquitous within the environment (Hazlett, 1990; Kiesecker et al., 1999; Ferrari, Wisenden & Chivers, 2010), as is the case for nitrogenous waste products that also act as disturbance cues (Hazlett, 1990; Kiesecker et al., 1999; Ferrari, Wisenden & Chivers, 2010; Brown et al., 2012). Therefore, the background level of cues could dictate how much cue is required to elicit a response. Although the background noise hypothesis relates to prey behavior, we show it may also apply to predators.

An alternative explanation for our results is that cortisol does not increase predatory behaviors per se, but that it elevated the endogenous cortisol levels in our perch due to uptake of the water-borne cortisol across the gills (Scott & Ellis, 2007). Glucocorticoids are known to rapidly affect activity patterns and aggressive behavior (Moore & Evans, 1999; Overli, Kotzian & Winberg, 2002; Chang et al., 2012). However, active perch moved around the tank a similar amount during each chemosensory test, and it was specifically inspection that were elevated in the cortisol tests of actively moving perch.

The behavioral response of perch to a cortisol stimulus suggests that the hormone could influence predatory behavior. Therefore, there is the intriguing potential of a chemosensory arms race between predator and prey. Populations of sticklebacks differ in their behavioral profiles due to predation pressure (Bell, 2005; Bell & Sih, 2007), and fish from high predation populations have been shown to have a lower cortisol release rate than those from low predation populations (Archard et al., 2012). This has been suggested to be beneficial as it could prevent excessive energy expenditure due to repeated stress responses caused by the presence of predators (Brown, Gardner & Braithwaite, 2005). However, a reduced stress response could have evolved due to predators selectively predating individuals that have a higher stress response.

Conclusions

Traditionally, research has focused on the function of the stress response experienced by prey animals during a predation event. However, in this study we show how an improved understanding of how predators respond to the stress responses of their prey could be useful when investigating the complexity of predator–prey interactions.

Supplemental Information

Data S1 Full dataset used for analysis

Click here for additional data file.

We thank Nick Beevers, David Fettes and Stuart Wilson for help with fish collection; Shaun Killen, John Laurie and Donald Reid for help with experiment logistics and Colin Adams and Neil Metcalfe for valuable advice. We thank Robert Burriss for comments on an earlier draft, and two anonymous reviewers for their constructive feedback.

Additional Information and Declarations

Competing Interests

Author Contributions

Animal Ethics

Field Study Permissions

Data Availability

The authors declare there are no competing interests.

Lindsay J. Henderson conceived and designed the experiments, analyzed the data, contributed reagents/materials/analysis tools, wrote the paper, prepared figures and/or tables, reviewed drafts of the paper.

Mary R. Ryan performed the experiments, reviewed drafts of the paper.

Hannah M. Rowland conceived and designed the experiments, contributed reagents/materials/analysis tools, wrote the paper, reviewed drafts of the paper.

The following information was supplied relating to ethical approvals (i.e., approving body and any reference numbers):

The study was performed under a UK Home Office licence (60/4292).

The following information was supplied relating to field study approvals (i.e., approving body and any reference numbers):

Permission was gained for catching perch from a privately owned trout fishery (Howend) and sticklebacks were collected from Balmaha pond after obtaining permission from the Scottish government.

The following information was supplied regarding data availability:

The raw data has been uploaded as Data S1.

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
