# Peer review of "Perch, Perca fluviatilis show a directional preference for, but do not increase attacks toward, prey in response to water-borne cortisol"

_PeerJ, doi:10.7717/peerj.3883_

## Round 0.1 · original submission · Major Revisions

Your manuscript has now been reviewed and the reviewer comments are appended below. You will see that, while they find your work of interest, they have raised points that need to be addressed by a major revision.

We therefore invite you to revise and resubmit your manuscript, taking into account the points raised.

Reviewer 1 ·

Basic reporting

The article is largely well-written, with sufficient references and background in most places. The raw data are available. I do not think that, given the controls in the experiment, that the hypotheses are truly being addressed as they are currently worded.

Experimental design

The research aims are within the scope of the journal. The research question is well defined, although I think more motivation is needed. I think that additional controls, such as behavior of a non-predatory fish, would have been useful. The ethics and methodological details are sufficient, though there are some things that should be clarified with the statistics.

Validity of the findings

There are some major statistical concerns (see general comments) that should be addressed. I think that, given that more or different controls would be useful for identifying if the change in behavior is indeed predatory, the discussion should be more speculative.

Additional comments

General comments

The manuscript is generally well written and clear. I think that it presents an interesting and relevant topic, as the idea that reduced stress responses may have evolved as a mechanism to avoid predators is an intriguing idea and worth exploring. However, I have some major concerns about the manuscript in its current state:

1. I have some major statistical concerns that I think need to be addressed to make the results more convincing.
a. You found that perch tended to show increased lunges toward stickleback during the cortisol treatment, but the majority of trials that were removed because the perch did not move or lunge at all were cortisol trials. It seems likely to me that removal of those cortisol trials (with zero lunges) may have biased the dataset, and it would be useful to run the stats with and without those trials. I understand that using those trials for proportion of time oriented or spent in a zone is biased by including them, but I don’t think that the same is true for the lunges.
b. I am somewhat confused by your statistical methods. I assume you ran 4 separate models, each with one fixed effect (treatment) and one random effect (perch ID)? If so, you should specify that in the table legend and replace “fixed effects” in the top row to “treatment”. Additionally, because the proportion of time spent in the chemical stimulus zone, proportion of time oriented in the chemical stimulus zone, and proportion of lunges are likely not independent, it would be more statistically rigorous to control for multiple tests.
c. I don’t understand why you used the proportion of lunges and not the total number of lunges. It seems to me that the total number of lunges would be a more accurate reflection of predatory behavior, and it would be useful to justify why proportions were used in this case. Further, how often did the perch lunge in the opposite stimulus zone and do you know what was the purpose of that behavior is?

2. Since you used stressed levels of cortisol (0.20), you need to justify why you used acclimated sticklebacks in your experiment or reframe your question. It seems to me that, to understand if cortisol itself is responsible for giving cues of prey vulnerability, a better control would be to compare cortisol alone (at stress induced levels) to visual cues of stressed prey and/or natural chemical cues of prey that are stressed.

3. Given that, as you state in the introduction, prey can respond to the cortisol emitted by conspecifics, how do you know that predators interpreted the cortisol as cues of prey and not as a signal of danger to themselves? Given that some predators froze, which is an antipredator behavior in many species, is it possible that this is the case? On the other hand, predators might be less willing to stay in the area with the cortisol if it was a cue for danger (the lunging results would also provide further proof of prey cues if the results still hold after redoing the statistics). I think it would have been extremely useful to have additional controls- to understand if a non-predatory fish reacts to the cortisol release in a different way or is this behavior specific to predators. I think that further discussion of this is necessary (and additional tests with this control, if possible, would definitely improve the manuscript). Suggestions for other possible controls are given below.

4. Clarification on what type of cortisol (free or conjugated) is dropped into the water would be useful, as receptors for conjugated steroids tend to be much more prevalent than receptors for free steroids and therefore, perch would have different abilities to detect free versus conjugated steroids.


Specific comments

Lines 30-32: Here and elsewhere, it feels like there are more commas than are needed in the text. Reducing the number of commas would improve the readability in sections.

Line 32: relative to presentations of natural chemical cues…

Line 33: In addition, perch tended to show increased…

Line 52: If you bring up the life and dinner principle, it would be useful to readers to define it here.

Line 74: Citation?

Lines 70-76: Here and elsewhere (e.g. lines 82-84), you repeatedly motivate the study by stating that we do not know that much about responses of predators to prey. There should be more motivation to do the study than the fact that it merely has not been done before, and indeed, I do think that it is an important topic of interest. I would remove these statements and instead be more specific in how “predators respond to the chemical cues released by prey would provide insight into the selective forces that shape the chemical ecology of predator-prey interactions”. For example, you allude generally to something in lines 371-372 of the discussion that is worth expanding on in the introduction.

Line 79: Such as? And given that there are lots of potential predatory cues, is it possible that putting any of them into the water at high concentrations would produce the same results? As in, I think it is worth discussing (and further investigating for someone) whether or not these results are associated with cortisol alone or if these results can be replicated by having high concentrations of any alarm pheromone in the water.

Line 98: influence

Line 112-113: I think part of this sentence is missing?

Line 113: Or cortisol might just provide information about location of prey, not necessarily vulnerability. I think a control where you run cortisol plus natural cues on one side, with just natural cues on the other, would differentiate between these.

Line 143: Did you reuse sticklebacks? If so, were there any notable changes in stickleback behavior over time? As in, sticklebacks could be responding behaviorally different to perch the first versus third time they see the predator.

Line 158: Again, what type of cortisol is important information.

Lines 160: Since that is not natural levels, but stressed levels of cortisol, I don’t understand why you used acclimated sticklebacks in your experiment, rather than stressed sticklebacks who would give visual cues of vulnerability and natural chemical cues of vulnerability. It seems to me that, to understand if cortisol itself is responsible for giving cues of prey vulnerability, that a better control would be to compare cortisol alone (at stress induced levels) to natural cues of prey that are stressed.

Lines 216-221: Again, why proportions? How often did they lunge not at stickleback?

Lines 224: Again, I assume that these are not independent. If they are not independent, controlling for multiple tests would be beneficial.

Line 228: Degrees of freedom are needed here and throughout

Line 234: Package?

Lines 251: Final sample sizes for each treatment after removals would be useful. Also, I think removing 12 tests from models looking at lunging behavior likely biases the results.

Lines 260: Including the statistics here and in the table is redundant- please choose one.

Lines 264-265: I’m not quite sure what you mean by “the chemical stimulus was dripped in the cortisol test relative to the water control”.

Lines 265-267: I may have missed this, but how did you measure movement? Just movement between zones?

Line 270: Range or standard error?

Lines 270-271: Again, this is a large difference, which makes me question why comparing unstressed visual and chemical cues to stressed cortisol levels helps you test “the hypothesis that the disturbance cue cortisol informs predatory behaviour when presented with prey.” I think you need to reframe and/or clarify your question, making it clear that you aren’t testing visual versus chemical cues of stress.

Lines 276-285: See my general comments- I think that alternative explanations for altered behavior should be considered here.

Lines 298-305: It is a little unclear how these sentences connect to the rest of the discussion- transitions would be useful

Line 307-308: But prey who have been repeatedly exposed to predators may have dampened cortisol responses

Line 316-317: I think that you can really only say this if you look at whether or not non-predatory fish behave in the same way.

Line 333-334: Affect behavior in what way? Also, “but” at the beginning of the next sentence might be replaced by “however”.

Line 335: that were elevated

Lines 337: What do you mean by undirected changes?

References: Make sure you italicize scientific names

Figure 2: I don’t understand what the numbers in the bars are- please include that in the figure legend. Also, it would be useful to see both sides of the standard error bars, especially since the data are not normally distributed and therefore, the error bars should not be symmetrical. 2C should have letter b instead of c over the cortisol treatment.

Reviewer 2 ·

Basic reporting

This is, in its majority, a very well written manuscript. It flows well and everything is backed up with what seems to be a very comprehensive literature review. The results obtained are relevant to the hypothesis and interpreted in a rigorous and informed way. In L335: change 'we elevated' to 'were elevated'.

I would encourage the authors to avoid using bar graphs and replace them with boxplots. This is because boxplots show the distribution of the raw data and, thus, are more informative.

Experimental design

The design is simple and well thought through. It is clear that the research question(s) fills a knowledge gap. Methods are described in detail. The passage were the hypotheses are stated, however, needs to re-worded. Paradoxically, this is the least clear paragraph in the whole manuscript.

Validity of the findings

Data analysis is mostly appropriate. Taking into account the 'repeated measures' nature of the experiment, the authors rightly included predator id as a random factor in their statistical models. However, I'd suggest they reanalyse their data regarding number of movements and number of attacks with a GLMM with a Poisson distribution, which tends to be the distribution that fits counts best.

The authors acknowledge the limitations of their study, and state the conclusions that derive strictly from the data presented.

Additional comments

I enjoyed reading the manuscript!

---

## Round 0.2 · Minor Revisions

One reviewer made some final observations regarding your manuscript. I recommend you to clarify the experimental details as requested by the referee. With these final improvements I will be able to accept your manuscript for publication.

Reviewer 1 ·

Basic reporting

It is well-written and clear, sufficient background and literature review.

Experimental design

Well-designed, research question clear and relevant, methods are sufficiently detailed.

Validity of the findings

Some minor statistical changes would be useful, but generally, the data are sound.

Additional comments

This version of the manuscript is greatly improved over the last version, both in terms of readability and in terms of robustness of the data. The abstract, introduction, and discussion all read very well, although some minor changes to the introduction and discussion (see below) would clear up a few remaining ambiguities. The additions to the methods were useful, although I still have some remaining statistical concerns.

Specific comments
Lines 60-62: It would be useful to give an example here of how this is the case.

Lines 67: information regarding the presence of predators?

Lines 81-82: Additional fitness benefits of releasing alarm cues? This paragraph is a bit unclear to me, as it starts with saying the predators use alarm cues to locate prey (which seems bad for the prey), but you then say that it is beneficial because it attracts more predators. Expanding and clarifying this paragraph would improve the readability.

Lines 83: Remove the comma between injury and attract

Lines 89-90: It would be useful to define disturbance cues here.

Lines 135-141: You make predictions here with respect to visual cues versus cortisol, but it would be useful to understand what differences you expect to see (if any) in natural cues versus cortisol.

Lines 263-264: I think it would be useful to clarify that these are (as I interpret it) lunges toward the stickleback on the side of the chemical stimulus versus lunges to sticklebacks on the visual cue only side.

Lines 276: Did you check to see if your data were overdispersed (i.e. if you should use Poisson or negative binomial distribution)?

Lines 272-275, 328-332: Does the total number of lunges and movements include the trials in which perch did not lunge? I agree that, in some cases, excluding individuals who do not react to the stimulus presented by the researchers is appropriate. For example, removing the samples in which the perch were immobile when trying to evaluate proportion of time spent in the chemical stimulus zone seems appropriate because, as you say, it is possible that they spent 100% in the chemical stimulus zone merely because they never moved. With the proportion of lunges, it is also appropriate. However, I do think that it is important to include those zeros in the trials examining total numbers of lunges/movements- if you remove them, you effectively remove many more zeros from the cortisol treatment group than you did from the other two treatment groups and this almost certainly biases the data.

As it is written, it is unclear if you did or did not include the trials with immobile perch in your current analysis, and you should show your data with and without those trials removed. In the minimum, you should show the additional analyses that you put in your letter “we do not think this is the case for our data because when we re-analyse the perch behaviour including only those perch who are active in all three conditions (9 perch, and 27 trials) our result is the same: Proportions: Duration, GLMM t = 3.62, P = 0.002; Orienting, GLMM t = 3.91, P < 0.001, Lunges, GLMM t = 4.34, P = 0.08. Total lunges: GLMM z = 4.50, P < 0.001.” I think this does provide support for the robustness of your results.

Lines 354-357- I would expand this a bit, or at least cite studies showing that immobility is a deterrent behavior.

Lines 360: Instead of “reduced predator avoidance”, I would say “a reduced ability to avoid predators”.

Line 366: I think a more specific transition here would make this a bit more readable. Instead of “Context can influence the behavioral response of fish to chemical cues”, it would be useful to say something along the lines of “Predators may respond differently to chemical cues of prey depending on the social or ecological context or depending on their own attributes” as the example you use primarily refers to differences in size/age in how perch respond to cues.

Lines 433-434: It would be useful to explain this. E.g. further tests with a non-predatory fish would be needed to confirm that changes in behavior are specific to predatory fish and not a general response to cortisol. I think that this would be interesting because it would likely confirm that this is indeed something specific to predator-prey interactions.

Reviewer 2 ·

Basic reporting

I have had a look at the response letter and the amended manuscript and I am happy to say that the authors did a great job with their revisions in agreement with my, and the other reviewer's, suggestions. Thus, I am happy for this MS to be accepted.

Experimental design

As I said in the previous review, I think the experimental design is well thought through.

Validity of the findings

Conclusions are backed up by the results of the experiment

Additional comments

The authors did a serious and thorough job with the revisions!

---

## Round 0.3 · accepted · Accept

I am pleased to inform you that your manuscript referenced above has been accepted for publication in PeerJ.